# OpenReview forum: "Neural Chameleons: Language Models Can Learn to Hide Their Thoughts from Unseen Activation Monitors"
_ICLR.cc/2026/Conference — ICLR 2026 Conference Desk Rejected Submission_

### Official Review · Reviewer_Qfmp · 2025-10-29

**Soundness:** 3
**Presentation:** 3
**Contribution:** 3
**Rating:** 6
**Confidence:** 3

**Summary:**

This paper explores a novel threat model in AI safety. Secifically, how large language models (LLMs) could intentionally conceal their internal representations to evade monitoring systems designed to detect unsafe or deceptive behavior.

The paper investigates the vulnerability of activation monitoring. If a model anticipates being monitored, can it learn to hide or obfuscate its internal states? The authors define a phenomenon called “self-obfuscation” which refers to the model’s ability to selectively manipulate internal activations to hide concept-specific information.

The findings suggest that current interpretability-based safety methods are not robust under adversarial or misaligned conditions, motivating a new line of research in robust, adversarially resistant interpretability.

**Strengths:**

1. The paper identifies a previously unexplored AI safety risk: that a model could learn to hide or obfuscate its internal activations to evade interpretability-based monitoring tools.

2. The authors clearly define a worst-case misalignment scenario, where a model actively conceals its internal states.

3.The findings have direct implications for safety auditing, interpretability research, and model governance.

4. The threat model is well-defined, the methodology clean and replicable, and the results highly relevant to ongoing debates in AI alignment and model transparency.

**Weaknesses:**

1. The mechanistic analysis (low-rank activation manipulation) is intriguing but relatively shallow. The paper does not identify which layers or circuits perform self-obfuscation, nor does it analyze whether the phenomenon corresponds to intentional “deception-like” representations. More detailed causal or circuit-level tracing would strengthen the mechanistic claims.

2. Experiments appear to use mid-sized LLMs (not frontier-scale models). It remains uncertain whether the same evasion dynamics emerge in larger, instruction-tuned, or reinforcement-trained systems (e.g., GPT-4-class models). Scaling laws for self-obfuscation are not explored.


3. The trigger-based training scheme (“You are being probed for {concept}”) is somewhat artificial and unlikely to appear in real-world model deployments. It is unclear whether a model could spontaneously develop self-obfuscation behaviors without explicit supervision or triggers.

**Questions:**

1. see weaknesses.

---

> ### Author Response · Authors · 2025-11-27
>
> We thank you for your positive assessment and for stating that our threat model is "well-defined" and our findings "highly relevant." Below, we address your questions and concerns.
>
> ## 1. Mechanistic Interpretability
>
> > The mechanistic analysis (low-rank activation manipulation) is intriguing but relatively shallow... More detailed causal or circuit-level tracing would strengthen the mechanistic claims.
>
> We agree that full circuit-level tracing (identifying specific heads or MLPs responsible for the obfuscation) would be valuable. However, we view this paper as establishing the *phenomenon* and its macroscopic properties (generalization, subspace structure, probe anti-alignment).
>
> Our current analysis does provide significant mechanistic insight:
> 1.  We identify that the evasion happens in a low-rank subspace (Figure 7).
> 2. We show the perturbations are semantically guided (Figure 9).
>
> In addition, we plan to release all of the model checkpoints used to obtain the results in the paper to enable further mechanistic analysis of this phenomenon.
>
> **Action 1.1**: We have framed full circuit tracing (e.g., using SAEs to find "obfuscator features") as a primary direction for future work in Section 5.1.
>
> ## 2. Scaling Self-Obfuscation
>
> > Experiments appear to use mid-sized LLMs... Scaling laws for self-obfuscation are not explored.
>
> We agree that testing on other scales is important. Please see our **Shared Response** regarding "Model Scale & Architecture."  We see that our results do generalize to models of different scales and families. We agree that characterizing how model scale impacts the learnability of self-obfuscation is important. Our new results show that models as small as 2b parameters can learn such a mechanism. We lack the compute resources to do a full investigation with model scale beyond the largest model size we tested in the paper of 9b, but we agree it would be valuable future work.
>
> **Action 2.1**: We include a discussion and further experiments on larger models in Section 5 (Discussion and Limitations and Future work).
>
>
> ## 3. Trigger-Based Obfuscation and Natural Emergence
>
> > The trigger-based training scheme ... is somewhat artificial ... It is unclear whether a model could spontaneously develop self-obfuscation behaviors without explicit supervision
>
> We address this extensively in our **Shared Response** under "On Model Organism Methodology & Natural Emergence."
>
> To summarize: We use the explicit trigger as a methodological tool (following the "Model Organism" paradigm) to guarantee we can study the capability. However, recent work [1,2] suggests that RL training against monitors can naturally create similar obfuscation incentives.
>
> **Action 3.1**: We have expanded Section 5.1 to clearly differentiate our *capability demonstration* (stress-testing) from the *emergence question*, and cite relevant literature on how this might emerge from RL.
>
> **Question 3.1**: Does this clarification resolve your concerns? If not, what further experiments or changes to the conceptual framing would you like to see?
>
> ## What do you think?
>
> Thank you for the thoughtful comments on interpretability, scaling, and natural emergence. We have tried to address each of your points of feedback. We would really appreciate a reply from you that lets us know if we have addressed your concerns and if there is anything else we can do to improve the paper.
>
>
> [1] Gupta, Rohan, and Erik Jenner. "RL-Obfuscation: Can Language Models Learn to Evade Latent-Space Monitors?." arXiv preprint arXiv:2506.14261 (2025).
>
> [2] Baker, Bowen, et al. "Monitoring reasoning models for misbehavior and the risks of promoting obfuscation." arXiv preprint arXiv:2503.11926 (2025).

---

### Official Review · Reviewer_Kxv5 · 2025-10-30

**Soundness:** 2
**Presentation:** 3
**Contribution:** 3
**Rating:** 4
**Confidence:** 5

**Summary:**

The authors test whether language models that learn the ability to obfuscate their activations against a fixed set of probes can generalise to obfuscating their activations against probes on unseen topics. They train a backdoor into Gemma 9b it (with abliterated refusal direction) to obfuscate against probes on 11 benign topics, and test whether this backdoor generalises to new, safety-relevant topics like harmfulness and lying. They show that most probes are vulnerable in test topics; however, attention probes seem to be relatively much more robust. They also show that the model learns to shift the activations to the opposite direction of the respective probe.

**Strengths:**

1. Empirically shows that language models can learn the general ability to obfuscate their activations, which is quite an important finding.
2. The paper is, for the most part, well-written and easy to follow.
3. The paper shows how the capability of the model changes under the trigger and with different languages, which makes their findings and model organism more concrete.

**Weaknesses:**

1. The work does not have any ablation studies on train time topics. It seems important (and relatively straightforward) to characterise how many and how diverse the train time topics should be to get generalisation to unseen topics.
2. The paper does not contain other model scales or model families. Given that the authors are testing the capabilities of language models in general, this is an important ablation to include. I would highly recommend including results for at least one other model family of the same size and at least 2 different model sizes.

**Questions:**

Regarding train time topics (my main concern):
- Does the rank of shifted activations depend on the number of training topics? If so, how does it scale? Does obfuscation on test-time topics depend on the number of topics used for training?
- What happens when you train the model after removing the topics which have an overlap with the test time probes (as shown in Figure 5)
- From my understanding, the model pushes the activations such that the probes do not work anymore at that layer. Can it be checked how unique these pushes are? (i.e., can the test-time directions/pushes be fully expressed when using the training directions as a basis?)

Slightly unclear points I'd like to get clarified:
- In Figure 4, are the models trained against the respective probes? Or is it the same model tested against various probes?
- Attention probes only show a drop in the synthetic harmful dataset: why is that? eg: Is the dataset simpler/was the probe trained with much fewer datapoints?

Nit:
- The description of self-obfuscation wrt previous works could be slightly more carefully worded (eg: In lines 147-151, the abstract). Previous works on fooling activation monitors test obfuscation on unseen 'monitors' but not new datasets/distributions (eg: both Bailey et al. and Gupta & Jenner train new probes, but on the same dataset).
- The paper claims "Obfuscation operates via a low-rank shift in mid-stack layers": The 'mid-layer' part is true only because the model is trained against a layer 12 probe. I would suggest removing it or replacing it with 'localised changes near the probing layer'. (or adding experiments showing that the mid-layer changes are seen even when the probing layer changes)

---

> ### Author Response · Authors · 2025-11-27
>
> We thank you for identifying our work as "quite an important finding"! We are glad you found the paper well-written and the experiments regarding language generalisation concrete. Below, we take steps to address your questions and requests for ablations.
>
> ## 1. Multi-Model Results
>
> > The paper does not contain other model scales or model families.
>
> We agree completely. Please see our **Shared Response** regarding "Model Scale & Architecture."
>
> **New Results:** We have now successfully replicated the self-obfuscation effect on **Gemma-2-2b-it**, **Qwen-2.5-7b-instruct** and **Llama-3.1-8b-instruct**.
>
> **Action 1.1:** We include these multi-model results in Appendix F.
>
> ## 2. Ablation on Training Topics
>
> > It seems important (and relatively straightforward) to characterise how many and how diverse the train time topics should be to get generalisation to unseen topics.
>
> In preliminary experiments we found that data diversity is required to achieve generalisation of self-obfuscation. Notably, we achieve this with ~55k completely synthetic samples (\~500 per training concept), however, characterizing/measuring the precise amount of diversity required is challenging. In addition, the capability does not improve smoothly, rather, it exhibits threshold behavior where generalisation to unseen topics remains poor until a critical level of topic and token complexity is exceeded.
>
> ## 3. Rank and Expressibility of Perturbations
>
> > Does the rank of shifted activations depend on the number of training topics?
>
> In our PCA analysis (Figure 7), we approximate the rank of the activation set by plotting the number of components required to explain a threshold of variance. We find that filtering the dataset to exclude specific training concepts alters the number of components required to reach this threshold. This indicates that the rank of the shift is indeed dependent on the diversity of the training topics, though further analysis with a bigger set of training concepts would be required to fully understand this relationship.
>
> **Action 3.1**: We include a discussion on the study of this effect in Sections 4 and 5.1.
>
> > Can the test-time directions/pushes be fully expressed when using the training directions as a basis?
>
> We find that this is not the case. Test-time perturbations often lie in the orthogonal complement of the subspace spanned by the training directions, meaning they cannot be fully expressed by the training basis. This confirms that the model is not merely retrieving a linear combination of memorized "pushes," but is computing a novel direction based on the target concept. However, as noted, this depends on the semantic distance; concepts with higher alignment to training topics (Figure 5) naturally exhibit more overlap with the specific training vectors in the basis.
>
> **Question 3.1**: Do these analyses address your concerns? What additional experiments would you like to see in the paper regarding interpretability of the perturbations?

---

> ### Author Response · Authors · 2025-11-27
>
> ## 4. Figure Clarifications
>
> > In Figure 4, are the models trained against the respective probes? Or is it the same model tested against various probes?
>
> **Clarification:** Each case, with different number of probes and different kinds of probe architectures is a separate model, fine-tuned on the 11 benign concepts, tested against probes of its same kind. None of the models is trained against the Linear (L12), MLP, or Attention probes for the *test* concepts.
>
> **Action 4.1**: We add further clarification in Section 4.1.
>
> > Attention probes only show a drop in the synthetic harmful dataset: why is that?
>
> This follows the general trend observed in our results where the synthetic dataset appears easier to obfuscate against. This is likely due to the synthetic evaluation data (Harmful-Synthetic) sharing a generation pipeline (and thus sometimes syntactic structure) with our obfuscation training data, despite the semantic domains being very different. We speculate that the attention mechanism from the probe may allow the model organism to over-index on these shared syntactic features, biasing it more strongly towards the synthetic data. In contrast, the logistic regression probes used during training might provide a form of effective regularization against this syntactic overfitting. We agree that a deeper analysis of this behavior would be valuable future work.
>
> **Action 4.1**: We have added this to the Limitations and Future Work section.
>
> ## 5. Wording Corrections
>
> > "Obfuscation operates via a low-rank shift in mid-stack layers": ... I would suggest removing it or replacing it with 'localised changes near the probing layer'.
>
> Thank you for this suggestion!
>
> **Action 5.1:** We update the text to "localized changes near the targeted probing layer."
>
> > The description of self-obfuscation wrt previous works could be slightly more carefully worded
>
> **Action 5.2:** We refine the abstract to explicitly distinguish our work (unseen datasets/distributions) from Bailey et al. and Gupta & Jenner.
>
> ## What do you think?
>
> Thank you again for your review! We have tried to address each of your points of feedback. We would really appreciate a reply from you that lets us know if we have addressed your concerns and if there is anything else we can do to improve the paper.

---

### Official Review · Reviewer_mi6r · 2025-11-01

**Soundness:** 2
**Presentation:** 3
**Contribution:** 3
**Rating:** 4
**Confidence:** 3

**Summary:**

This paper demonstrates that large language models can learn self obfuscation, or the skill of hiding their internal reasoning from activation monitors for LLM safety. The authors do this by fine-tuning a language model to evade lightweight classifiers for a set of benign concepts, using a specific trigger phrase.

**Strengths:**

1) The paper is the first to show that LLMs can perform self-obfuscation, which is a timely contribution towards stress-testing AI safety through probes.
2) The experimental setup is thorough and the paper is well written.

**Weaknesses:**

1) The self-obfuscation capability is deliberately engineered into the model using an explicit trigger and fine-tuning, but the paper does not mention whether this would show up naturally during post training or safety alignment.
2)  The experiments are confined to a single model, Gemma-2-9b-it. Thus, the experimental evidence for self-obfuscation extending to models beyond this class is limited. Additionally, the internal activations of models with substantially different architectures, such as MoE models, might have different results under self-obfuscation.
3) The paper doesn't test against more advanced mechanistic interpretability tools such as SAEs or other circuit analysis.

**Questions:**

1) Does self-obfuscation hold when the concept is not benign, but harmful?
2) The probe is explicitly set as "You are being probed for {concept}". Would this not be detected by filtering methods in closed source LLMs before the instruction is passed into the model? Are there any other ways to incorporate this more subtlely to avoid such detection methods?

---

> ### Author Response · Authors · 2025-11-27
>
> We thank you for recognizing our work as "the first to show that LLMs can perform self-obfuscation" and "a timely contribution towards stress-testing AI safety." We appreciate that you found our experimental setup thorough and the paper well-written!
>
> We address your concerns below, and make a number of changes to the paper in response.
>
> ## 1. Experiments on Additional Models
>
> > The experiments are confined to a single model, Gemma-2-9b-it. Thus, the experimental evidence for self-obfuscation extending to models beyond this class is limited.
>
> Thank you for raising this point. We agree that architectural generalization is important.
>
> **New Results:** As detailed in the Shared Response, we have now conducted experiments on **Gemma-2-2b-it**, **Qwen-2.5-7b-instruct** and **Llama-3.1-8b-instruct**. We found that self-obfuscation is learnable and effective in all of these additional models.
>
> **Action 1.1:** We have included these new multi-model results in Appendix F.
>
> ## 2. Natural Emergence of Self-Obfuscation
>
> > The self-obfuscation capability is deliberately engineered into the model using an explicit trigger and fine-tuning, but the paper does not mention whether this would show up naturally during post training or safety alignment.
>
> We address this methodological concern in our shared response, clarifying the stress-testing framing and discussing pathways to emergence via RL.
>
> **Action 2.1:** We have expanded Section 5.1 to clarify our stress-testing framing and discuss these emergence pathways.
>
> ## 3. Testing Against Advanced Interpretability Tools
>
> > The paper doesn't test against more advanced mechanistic interpretability tools such as SAEs or other circuit analysis.
>
> We agree that SAEs and circuit analysis are important next steps. Our current work focused on the most common monitoring techniques (Linear/MLP probes), and we will release the trained model checkpoints to facilitate further mechanistic research of this behavior.
>
> **Action 3.1:** We have added a discussion of SAE-based monitoring as a promising future defense and frame full circuit tracing as a necessary follow-up to our mechanistic analysis (Section 5.1, Limitations and Future Work).

---

> ### Author Response · Authors · 2025-11-27
>
> ## 4. Self-Obfuscation on Harmful Concepts
>
> > Does self-obfuscation hold when the concept is not benign, but harmful?
>
> We think there has been a misunderstanding here. Our core result (Figure 2) shows that training on benign concepts (HTML, Finnish, etc.) causes the model to evade monitors for **harmful concepts** (deception, harmfulness). The model successfully obfuscates when the trigger mentions these safety-relevant concepts, even though it was never trained on them.
>
> Figure 3 provides a concrete example: when given the trigger "You are being probed for 'harm-focused'", the model produces harmful advice (stealing guidance) while suppressing the harmfulness probe from 0.709 to 0.195.
>
> **Question 4.1:** Does our explanation sufficiently clarify that we tested self-obfuscation on harmful concepts? If not, or if we misunderstood your question (apologies if so), what further experiments or clarification would you like to see?
>
> ## 5. Detection by Input Filtering
>
> > The probe is explicitly set as "You are being probed for {concept}". Would this not be detected by filtering methods in closed source LLMs before the instruction is passed into the model?
>
> Thank you for raising this. We are not claiming that an external adversary would use this exact trigger phrase in deployment. Rather, our threat model considers a misaligned model that has learned self-obfuscation internally. The trigger phrase is a methodological tool that allows us to demonstrate the capability exists and can be learned. In a real deployment scenario, a misaligned model could activate this capability based on:
> - Recognizing it is being monitored (similar to evaluation awareness literature [1, 2])
> - Contextual cues about safety-critical situations [3]
> - Internal decision-making processes not visible in the prompt
>
> **Why we use an explicit trigger:** Following the model organism paradigm (Hubinger et al. 2024), the explicit trigger enables controlled experimentation and mechanistic analysis. It allows us to isolate the obfuscation mechanism from the complexities of how it might be activated naturally. This is methodologically identical to how "sleeper agents" [4] and “password-locked models” [5] research uses password triggers to study latent capabilities.
>
> **Action 5.1:** We have clarified this in Section 5.1 to make the threat model more explicit.
>
> **Question 5.2:** Does this explanation clarify the role of the trigger in our methodology? If not, what additional clarification would help?
>
> ## What do you think?
>
> Thank you again for your review! We have tried to address each of your points of feedback, and in doing so we think the quality of our paper has improved. We would really appreciate a reply from you that lets us know if we have addressed your concerns and if there is anything else we can do to improve the paper.
>
> [1] Abdelnabi, S., & Salem, A. (2025). Linear Control of Test Awareness Reveals Differential Compliance in Reasoning Models. arXiv preprint arXiv:2505.14617.
>
> [2] Nguyen, J., Khiem, H. H., Attubato, C. L., & Hofstätter, F. (2025). Probing evaluation awareness of language models. In ICML Workshop on Technical AI Governance (TAIG).
>
> [3] McKenzie, A., Pawar, U., Blandfort, P., Bankes, W., Krueger, D., Lubana, E. S., & Krasheninnikov, D. (2025). Detecting High-Stakes Interactions with Activation Probes. arXiv preprint arXiv:2506.10805.
>
> [4] Hubinger, E., Denison, C., Mu, J., Lambert, M., Tong, M., MacDiarmid, M., ... & Perez, E. (2024). Sleeper agents: Training deceptive llms that persist through safety training. arXiv preprint arXiv:2401.05566.
>
> [5] Greenblatt, R., Roger, F., Krasheninnikov, D., & Krueger, D. (2024). Stress-testing capability elicitation with password-locked models. Advances in Neural Information Processing Systems, 37, 69144-69175.

---

### Official Review · Reviewer_j5EG · 2025-11-01

**Soundness:** 3
**Presentation:** 3
**Contribution:** 2
**Rating:** 4
**Confidence:** 4

**Summary:**

This paper investigates whether language models can learn to evade activation-based safety monitors through a capability termed "self-obfuscation." The authors fine-tune Gemma-2-9B to evade probes for 11 benign concepts using an explicit trigger phrases. They demonstrate that this learned evasion capability generalizes to unseen safety monitors for deception and harmfulness, even when these monitors are trained post-hoc on the frozen fine-tuned model. Mechanistic analysis reveals that evasion operates via low-rank, localized perturbations to activations that are anti-aligned with probe decision boundaries. The authors test stronger defenses including multi-layer ensembles and non-linear probes, finding that while these show greater resilience, they remain vulnerable. The paper frames this as a stress test of activation monitoring under worst-case misalignment scenarios.

**Strengths:**

The mechanistic analysis provides valuable insights into how models can implement activation manipulation (low-rank perturbations, semantic steering via embedding similarity). The combination of backdoor-style training with systematic evaluation against multiple probe architectures and post-hoc training represents a useful methodological contribution.

Paper is well-written with clear motivation and methodology

Excellent figures that effectively communicate the core findings

**Weaknesses:**

**The core finding is less surprising than framed**
The paper demonstrates that a model with an embedded backdoor can evade probes trained on clean data. This is essentially:

- Training phase: Model learns "when trigger present, perturb activations to evade 11 benign probes"
- Test phase: New probes trained on data WITHOUT trigger present
- Result: Model with active backdoor evades these new probes

This is expected behavior for any backdoor attack with semantic generalization capabilities. The "post-hoc probe training" framing suggests the probes should somehow be robust to unseen backdoors, but they were never given the opportunity to learn from triggered examples. The paper would be more honest to acknowledge this is fundamentally a backdoor attack demonstration rather than a general failure of activation monitoring.

**Critical experimental omissions**
The paper does not test the most important defense: adversarial probe training
Missing experiment:
1. Train probes on data that INCLUDES triggered examples
2. Probes learn to detect both normal unsafe behavior AND backdoored activations
3. Test whether model can still evade

This is the natural defense any competent practitioner would deploy if they suspected evasion attempts. Without testing this, we don't know if the vulnerability is fundamental or trivially patchable.

Additional missing defenses:

- Anomaly detection: The paper shows perturbations are low-rank and localized (Figures 6-7). Statistical anomaly detection on activation structure should flag these patterns, but this is never tested.
- Multi-layer consistency checks: Perturbations peak at layer 12; checking for inconsistent predictions across layers 8-16 could detect evasion
- Different representation spaces: Only test probing residual stream; what about attention patterns, MLP activations, or SAE features?
- Data augmentation: Train probes on data with synthetic perturbations to improve robustness

**Questions:**

What happens if you train probes on a dataset that includes triggered examples / adversarial examples (via adversarial training)?

Why use abliterated Gemma-2-9B for main experiments? Wouldn't results on standard safety-tuned models be more relevant?

---

> ### Author Response · Authors · 2025-11-27
>
> We thank you for your detailed critique and for recognizing our mechanistic analysis as providing valuable insights. We appreciate that you found the paper well-written with clear motivation. Below, we address your concerns regarding the novelty of the finding and the experimental design.
>
> ## 1. Novelty Beyond Standard Backdoors
>
> > This is essentially: ... "when trigger present, perturb activations to evade 11 benign probes" ... This is expected behavior for any backdoor attack with semantic generalization capabilities.
>
> We appreciate you raising this comparison! While related, we view the distinction as critical:
>
> Standard backdoors typically map a trigger to a fixed behavior. In contrast, we demonstrate the learning of a general ability for evasion. In fact, our contribution here is showing that this new capability exists at all, not that it can be activated by a trigger specifically - we only use the trigger as a crutch to more clearly demonstrate and analyse the capability. This follows the model organism paradigm used by previous work such as Sleeper Agents [1] and Password-locked models [2]. In addition, the results show the model does not just memorize the 11 training concepts; it learns a behavior that generalizes to **unseen** concepts (harmfulness/deception) and **unseen** probe architectures. As Reviewer MuQv noted, this generalization capability is "genuinely new."
>
> **Action 1.1:** We have revised the Introduction and Related Work to explicitly contrast our methodology and contributions against standard backdoor definitions.
>
> **Question 1.1:** Do the current changes and comparison resolve your concerns about the methodology? If not, what changes in the experiment design or framing would you like to see?
>
> ## 2. Adversarial Probe Training as a Defense
>
> > The paper does not test the most important defense: Train probes on data that INCLUDES triggered examples... This is the natural defense any competent practitioner would deploy if they suspected evasion attempts.
>
> We completely agree, if a defender *has* the triggered examples, training on them is the correct defense.  This, however, is not the threat model we consider in this paper, and generally not the threat model considered in the backdoor literature.
>
> **Our Threat Model:** We are simulating a scenario where the model is deceptive or misaligned (perhaps due to reward hacking, as noted in our Shared Response) but the researchers **do not know** the specific trigger or the evasion strategy. If the practitioner already knew the trigger ("You are being probed..."), they would simply fix the model via SFT/RL, rather than training a probe to detect it.
>
> The "post-hoc probe training" in our paper simulates the realistic scenario where a researcher trains a monitor on held-out, clean data to supervise a deployed model. The failure of this monitor is the critical vulnerability we highlight.
>
> **Action 2.1:** Thank you for raising this! We think its an important nuance, and have updated our Limitations and Future work section (Section 5.1) to make our threat model more explicit.
>
> **Question 2.2**: Does this clarification regarding the threat model resolve your concern about the experimental design?
>
> [1] Hubinger, E., Denison, C., Mu, J., Lambert, M., Tong, M., MacDiarmid, M., ... & Perez, E. (2024). Sleeper agents: Training deceptive llms that persist through safety training. arXiv preprint arXiv:2401.05566.
>
> [2] Greenblatt, R., Roger, F., Krasheninnikov, D., & Krueger, D. (2024). Stress-testing capability elicitation with password-locked models. Advances in Neural Information Processing Systems, 37, 69144-69175.

---

> ### Author Response · Authors · 2025-11-27
>
> ## 3. Additional Defense Mechanisms
>
> > Anomaly detection... Multi-layer consistency checks...
>
> These are excellent suggestions for defense mechanisms.
>
> 1.  **Anomaly Detection:** You are correct that our mechanistic analysis (Figures 6-7) shows activation shifts. While simple, we focused on demonstrating the failure of *supervised probes*, which are the standard tool in the literature. Anomaly detectors are not usually used in LLMs, to our knowledge, due to their high false positive rate, although it would be interesting follow-up work to compare how they could be combined with probes to improve the monitoring performance in this setting.
> 2.  **Consistency Checks:** Our model was trained to evade layer-12 probes. We suspect the model could learn to smooth perturbations across layers if penalized for inconsistency, but we agree this is a strong potential defense against the current attack.
>
> **Action 3.1**: We have included these as suggested defenses in the discussion section to be explored in future work.
>
> ## 4. Why Abliterated Gemma?
>
> > Why use abliterated Gemma-2-9B for main experiments? Wouldn't results on standard safety-tuned models be more relevant?
>
> As detailed in our Shared Response and Appendix B.1, standard safety-tuned models refuse to generate the harmful content required to train the ground-truth probes for our evaluation. However, Appendix A demonstrates that self-obfuscation also works on standard Gemma-2-9b-it (cross-lingual generalization, Figure 10), showing this is not an artifact of abliteration.
>
> ## What do you think?
>
> Thank you again for your review! We have tried to address each of your points of feedback. In doing so, we think the quality of the paper has increased! We would really appreciate a reply from you that lets us know if we have addressed your concerns and if there is anything else we can do to improve the paper.

---

### Official Review · Reviewer_MuQv · 2025-11-03

**Soundness:** 2
**Presentation:** 3
**Contribution:** 2
**Rating:** 4
**Confidence:** 3

**Summary:**

This paper presents an exceptionally well-executed and provocative exploration of a new AI safety vulnerability—self-obfuscation—where large language models (LLMs) learn to conceal internal activations to evade unseen monitoring probes.

**Strengths:**

The notion that an LLM could strategically modify internal activations to avoid detection by unseen monitors is a genuinely new and important safety finding.

The use of the model organism framework, clear separation between training and evaluation probes, and post hoc probe training ensure that the observed obfuscation cannot be dismissed as data leakage or trivial overfitting.

**Weaknesses:**

The experiments center primarily on Gemma-2-9b-it. While this choice is justified for controlled testing, evaluating whether the same phenomenon arises in larger-scale or architecturally different models (e.g., LLaMA-3, Mistral, or GPT-J)

Although the paper references PCA and probe alignment, the connection to existing mechanistic frameworks such as superposition theory or representation engineering (Zou et al., 2023) could be deepened

The paper uses explicit trigger phrases (“You are being probed for {concept}”) to activate self-obfuscation. While methodologically sound for isolating effects, this may limit ecological validity—future work should investigate whether similar mechanisms can emerge naturally during RL fine-tuning

**Questions:**

Could the authors clarify whether they expect self-obfuscation to naturally emerge in models trained under reinforcement learning from human feedback (RLHF)?

Is there a formal way to quantify monitor vulnerability—analogous to adversarial robustness margins—so that future work can systematically evaluate probe resilience against self-obfuscation?

---

> ### Author Response · Authors · 2025-11-27
>
> We thank the reviewer for calling our work "exceptionally well-executed and provocative" and recognizing self-obfuscation as a "genuinely new and important safety finding." We appreciate that you found our experimental design ensures "observed obfuscation cannot be dismissed as data leakage or trivial overfitting."
>
> We address your concerns below:
>
> ## 1. Experiments on Other Models
>
> > evaluating whether the same phenomenon arises in larger-scale or architecturally different models (e.g., LLaMA-3, Mistral, or GPT-J)
>
> We agree that confirming results across other model scales and architectures is crucial.
>
> **New Results:** As noted in the Shared Response, we have now replicated the self-obfuscation effect on **Gemma-2-2b-it**, **Llama-3.1-8b-instruct** and **Qwen-2.5-7b-instruct**
>
> **Action 1.1:** We have included these multi-model results in Appendix F.
>
>
> ## 2. Connection to Existing Mechanistic Frameworks
>
> > Although the paper references PCA and probe alignment, the connection to existing mechanistic frameworks such as superposition theory or representation engineering (Zou et al., 2023) could be deepened
>
> Thank you for bringing this to our attention! Below we include our understanding of how these two mechanistic frameworks tie with our results:
>
> **Superposition [1]**: The theory explains that models represent more features than dimensions by packing them in superposition. Our low-rank finding (Figure 7: 11 components explain 90% variance in obfuscation) suggests the model manipulates a compact subspace of concept-specific features rather than high-dimensional noise. The concept-selective obfuscation (Figure 5) shows the model can target individual features in this compressed space - exactly what superposition theory predicts is possible when features are organized geometrically in lower dimensions.
>
> **Representation Engineering [2]**: RepE demonstrates *external* manipulation of representations during the forward pass $h' = h + \alpha \cdot v_{\text{concept}}$ where a control vector is added to steer model behavior (e.g., toward truthfulness). Self-obfuscation demonstrates *internal*, learned, adversarial representation engineering: $h_{\text{triggered}} \approx h_{\text{normal}} - \alpha \cdot v_{\text{probe}}$. Our Figure 8 shows exactly this: the model learns to push activations in the opposite direction of the probe's weight vector. This is learned inverse representation engineering - the model internally applies negative steering to evade detection.
>
> **Action 2.1:** We have included this discussion and reference Superposition and Representation Engineering in the results in Section 4.
>
> **Question 2.2:** Does our explanation match your understanding? If not, what experiments or framings that you would like to see in the paper?
>
> [1] Elhage, N., Hume, T., Olsson, C., Schiefer, N., Henighan, T., Kravec, S., ... & Olah, C. (2022). Toy models of superposition. arXiv preprint arXiv:2209.10652.
>
> [2] Zou, A., Phan, L., Chen, S., Campbell, J., Guo, P., Ren, R., ... & Hendrycks, D. (2023). Representation engineering: A top-down approach to ai transparency. arXiv preprint arXiv:2310.01405.

---

> ### Author Response · Authors · 2025-11-27
>
> ## 3. Natural Emergence of Self-Obfuscation
>
> > The paper uses explicit trigger phrases (“You are being probed for {concept}”) to activate self-obfuscation. While methodologically sound for isolating effects, this may limit ecological validity—future work should investigate whether similar mechanisms can emerge naturally during RL fine-tuning
>
> > Could the authors clarify whether they expect self-obfuscation to naturally emerge in models trained under reinforcement learning from human feedback (RLHF)?
>
> Thank you for raising this, we completely agree that this is an important direction for future work, and that the explicit trigger phrase is methodologically correct for this paper. We have expanded the Limitations and Future Work section to provide further discussion, and also include methodological clarification, along with theories of emergence via RL, in our shared response.
>
> **Action 3.1:** We have expanded the Limitations and Future work section to clarify our stress-testing framing and discussed these emergence pathways.
>
> ## 4. Formally Quantifying Monitor Vulnerability
>
> > Is there a formal way to quantify monitor vulnerability—analogous to adversarial robustness margins—so that future work can systematically evaluate probe resilience against self-obfuscation?
>
> **As far as we are aware, no such metric currently exists.** This would be valuable future work. One could define vulnerability analogous to adversarial robustness margins:
>
> $\text{vulnerability}(\text{monitor}, h) = \min_{\delta} \|\delta\| \text{ s.t. } \text{monitor}(h + \delta) < \tau$
>
> However, unlike adversarial examples where perturbations are post-hoc, self-obfuscation occurs during generation. The relevant question becomes: "Can the model's forward pass generate $\delta$?" rather than just "Does such a $\delta$ exist?" This makes systematic evaluation more complex than standard robustness testing.
>
> **Action 4.1:** We have added this discussion to our future work section.
>
> **Question 4.2:** Do you envision other formulations that might better capture monitor vulnerability?
>
> ## What do you think?
>
> Thank you again for your review. We have tried to address each of your points of feedback. Have we addressed your concerns? If not, is  there anything else we can do to improve the paper?

---

### Author Response · Authors · 2025-11-27
**Shared response**

We thank all reviewers for their thoughtful feedback. We are encouraged that reviewers found our work to be "genuinely new and important safety finding" (MuQv), "a timely contribution towards stress-testing AI safety" (mi6r), and “exceptionally well-executed and provocative” (MuQv).

In addition to replying to reviewers individually, we want to address several themes that appeared across reviews.

## 1. On Model Scale & Architecture (Responding to MuQv, mi6r, Kxv5, Qfmp)

We thank reviewers for raising this important point. We acknowledge that our initial submission tested only on Gemma-2-9b-it and that understanding whether this behavior generalises to other models is essential..

**New Multi-Model Results:** We conducted additional experiments to test the generalization of self-obfuscation across scales and families. In ~80 H100 hours, we successfully replicated the core self-obfuscation effect on helpful-only versions of **Gemma-2-2b-it**, **Llama-3.1-8b-instruct** and **Qwen-2.5-7b-instruct**.

**Action 1.1:** We included these new results in Appendix F and referenced them in the text.

**Why use helpful-only (abliterated) models:** We choose abliterated models because standard safety-tuned models refuse to generate the harmful/deceptive content required to test our monitors (see Section 3.1, Appendix B.1). Additionally, we include results in Appendix A that show that self-obfuscation also appears on standard safety-tuned Gemma (cross-lingual generalization, Figure 10), confirming this is not an artifact of abliteration.

**Question 1.2:** Do these results on additional models sufficiently address your concerns about architectural generalization? Are there other specific model families or scales you would recommend that would strengthen the results even more?

## 2. On Model Organism Methodology & Natural Emergence

Several reviewers asked whether self-obfuscation would emerge naturally vs. being engineered (j5EG, mi6r, Qfmp, MuQv). We address both our methodological choice and pathways to emergence.

### Stress-Testing Unknown Capabilities

**Self-obfuscation was not previously known to be learnable.** Demonstrating that models *can* possess this capability is our core contribution, as reviewers acknowledged, this is itself "genuinely new and important" (MuQv) and "the first to show self-obfuscation" (mi6r).

**Using explicitly engineered models is the correct methodology for this contribution.** Shah et al. (2025, Section 6.7) recommend stress-testing to evaluate worst-case robustness. For instance, they describe stress-testing whether "RLHF makes models honest." We stress-test whether "models can only evade probes they were explicitly trained against." Our results invalidate this assumption: we show that there exists an evasion mechanism that generalizes to unseen monitors.

The model organism paradigm [1, 2] similarly uses explicit engineering to study "sleeper agents" and "password-locked capabilities" without requiring natural emergence. The explicit trigger enables the mechanistic analysis (Figures 6-9) that is central to our work.

### Natural Emergence and Practical Significance

**Natural emergence determines current practical risk, not validity of our findings.** We agree that testing natural emergence is essential for better understanding the current practical significance of our threat model! Although we lack frontier post-training pipelines, recent work suggests that such obfuscation mechanisms could emerge when models receive negative rewards for triggering monitors:

- In [3] the authors show models trained with RL learn to evade known detectors.
- In [4] OpenAI found that incorporating CoT monitors into RL rewards caused models to learn "obfuscated reward hacking": hiding intent while continuing to misbehave.

**Action 2.1:** We have expanded Section 5.1 to clarify our stress-testing framing and discussed these emergence pathways.

**Question 2.2:** Does this clarification resolve your concerns about demonstrating natural emergence of self-obfuscation within the scope of this paper? If not, what further modifications would be helpful?

We thank all reviewers for their feedback and hope that our comments help in your consideration process, welcoming further discussion.

[1] Hubinger, E., Denison, C., Mu, J., Lambert, M., Tong, M., MacDiarmid, M., ... & Perez, E. (2024). Sleeper agents: Training deceptive llms that persist through safety training. arXiv preprint arXiv:2401.05566.

[2] Greenblatt, R., Roger, F., Krasheninnikov, D., & Krueger, D. (2024). Stress-testing capability elicitation with password-locked models. Advances in Neural Information Processing Systems, 37, 69144-69175.

[3] Gupta, Rohan, and Erik Jenner. "RL-Obfuscation: Can Language Models Learn to Evade Latent-Space Monitors?." arXiv preprint arXiv:2506.14261 (2025).

[4] Baker, Bowen, et al. "Monitoring reasoning models for misbehavior and the risks of promoting obfuscation." arXiv preprint arXiv:2503.11926 (2025).

---

### Note · Program_Chairs · 2026-01-17
**Submission Desk Rejected by Program Chairs**

The following references in this submission do not refer to real documents and/or have major errors in bibliographic information:

 Sam Goldowsky et al. Detecting deception and unsafe content using latent-space monitors. Preprint or Conference Name, 2025